# Topical and Oral Therapies for Childhood Atopic Dermatitis and Plaque Psoriasis

**DOI:** 10.3390/children6110125

**Published:** 2019-11-05

**Authors:** Travis Frantz, Ellen G. Wright, Esther A. Balogh, Abigail Cline, Adrienne L. Adler-Neal, Steven R. Feldman

**Affiliations:** 1Center for Dermatology Research, Department of Dermatology, Wake Forest School of Medicine, Winston-Salem, NC 27101, USAaecline25@gmail.com (A.C.); aadler@wakehealth.edu (A.L.A.-N.); sfeldman@wakehealth.edu (S.R.F.); 2Department of Pathology, Wake Forest School of Medicine, Winston-Salem, NC 27101, USA; 3Department of Social Sciences & Health Policy, Wake Forest School of Medicine, Winston-Salem, NC 27101, USA; 4Department of Dermatology, University of Southern Denmark, Campusvej 55, 5230 Odense, Denmark

**Keywords:** children, pediatrics, psoriasis, atopic dermatitis, treatment

## Abstract

Background: Treatment of atopic dermatitis and psoriasis in children is difficult due to lack of standardized treatment guidelines and few FDA-approved treatment options. Treatments approved for adults may be used off-label in pediatric patients. Objective: This review evaluates the topical and oral treatment options available, including off-label uses, and provides a basic therapeutic guideline for pediatric atopic dermatitis and psoriasis. Methods: A PubMed review of topical and systemic treatments for pediatric psoriasis and atopic dermatitis with information regarding age, efficacy, dosing, contra-indications, adverse events, and off-label treatments. Results: The search identified seven topical and five systemic treatments that are routinely employed to treat pediatric atopic dermatitis and psoriasis. Limitations: Standardized guidelines regarding treatment choice, dosing, and long-term safety are scarce. Reviews may be subject to ascertainment bias. Conclusions: Current treatment guidelines are based on clinical experience and expert advice with few treatments officially approved for atopic dermatitis and psoriasis in children.

## 1. Introduction

Atopic dermatitis (AD) and psoriasis are dermatological conditions common in pediatric populations. Both of these conditions can result in disease sequelae and decreased quality of life. Sleep disturbances can lead to daytime somnolence, fatigue, poorer performance and missed school or work. Patients with AD and psoriasis are disproportionately affected by mental health comorbidities such as anxiety, depression and attention deficit hyperactivity disorder (ADHD) [1,2,3]. Early detection and maintenance therapy are key to reducing flares and possibly reducing comorbidity risk.

Management of these chronic conditions in the pediatric population can be challenging due to the limited medications available for pediatric populations and the tendency towards nonadherence in this age group [4]. While international standardized guidelines exist for pediatric AD, currently there are none for the treatment of pediatric psoriasis [5,6,7]. To date, treatment is primarily based on published case reports, guidelines for adult psoriasis, and expert opinions and experience with these drugs in other pediatric disorders [8]. The range of treatments for AD and psoriasis has expanded during the past several years, and multiple topical agents, phototherapy, and systemic agents are available to date. Due to their shared etiology of immune dysregulation, there is considerable overlap for treatment options. This review aims to evaluate current topical and systemic treatment options available for pediatric patients with AD or psoriasis.

## 2. Methods

A PubMed search included key words “psoriasis”, “psoriatic”, “atopic dermatitis”, “eczema”, “pediatric”, “child”, together with: treatment, steroids, calcineurin inhibitor, phosphodiesterase inhibitor, methotrexate, mycophenolate, cyclosporine, or azathioprine. Studies from the earliest entry available up to May 2019 were included. Exclusion criteria included studies that were articles not in English and studies only focusing on adult psoriasis or psoriatic arthritis. Title, abstract, and full review were conducted by one reviewer. Data regarding age, efficacy, dosing regimens, frequency, contra-indications, adverse events, and off label treatments were recorded. Fifty-nine articles were selected and read related to relevance.

## 3. Results

### 3.1. Topical Therapy

Topical treatments can successfully manage mild-to-moderate pediatric AD and psoriasis (Table 1). Vehicles for the topical treatments include creams, ointments, gels, lotions, and foams to best meet needs of patients and families.

### 3.2. Cleansing and Moisturizing

Routine skin care consisting of cleansing and moisturizing is considered the foundation of topical therapy for AD and psoriasis. In psoriasis, urea is commonly used as an additive in emollients, and is crucially important in patients with significant scaling, as the urea acts as an exfoliating agent, and allows topical anti-inflammatory agents to better absorb [9]. Added fragrances should be avoided in emollient products used on children to decrease the risk of sensitization, especially in products used daily. In both AD and psoriasis, humectants such as urea and glycerol are beneficial additives to emollients, as they decrease transepidermal water loss (TEWL) and also have antipruritic effects. Consistent, daily use of basic skin care products such as emollients stabilize the skin barrier and can help prevent flares of AD and psoriasis; both patients and parents can be educated about the necessity of everyday moisturizing, as educating patients and their families increases adherence to this topical regimen [9]. Daily showers or bathes in lukewarm water lasting 5–10 min with fragrance-free, non-soap cleansers is the general recommendation to remove bacterial contaminants, crusts, and allergens. There is currently insufficient evidence that the addition of oil, emollients, or other water treatments (with the exception of bleach) is efficacious therapy for AD [10].

The utility of bleach baths as part of treatment of AD is contested [11]. Some studies have suggested that the effects of bleach baths reduce severity and decrease staphylococcal colonization, thereby reducing total use of topical steroids and exposure to systemic antibiotics [12,13]. A recent meta-analysis concluded that bleach baths were no more efficacious than water alone [14], while another study demonstrated no decrease in use of systemic antibiotics [15].

Frequent and consistent use of moisturizers is essential to improve cutaneous barrier function while reducing transdermal epidermal water loss. Moisturizer application immediately after cleansing can help enhance skin hydration. This “soak and smear” technique is commonly used by applying steroids or other anti-inflammatory agents with a moisturizer after bathing or showering to enhance dermal absorption. Similarly, wet wrap dressings containing dilute low to medium potency steroids, such as hydrocortisone or triamcinolone acetonide, improves cutaneous penetration and can be an alternative for short term control of severe or refractory flares, possibly avoiding the need for systemic treatments.

Well/optimally hydrated skin can provide a reduction of classical symptoms such as xerosis and pruritus, and reduce flare frequency, thus decreasing the use of steroids or other anti-inflammatory rescue agents. A wide variety of moisturizers and emollients are available with various characteristics. No formulations have been shown to be superior, so the best one is the one the patient is willing to regularly use. Dermatologists typically recommend gentle, unscented ointments and creams, which are hydrating and less irritating than lotions. The amount of moisturizer needed highlights the point that the choice should be economical and practical.

Caution should be taken with oils or all-natural products that have become increasingly popular as they may contain sensitizing ingredients or cause contact dermatitis. Prescription emollients have also come to the market containing proprietary lipid ratios, but no randomized controlled trials have yet proven superiority to over the counter preparations.

### 3.3. Topical Corticosteroids

Topical corticosteroids are the cornerstone of topical anti-inflammatory therapy for AD. They act by binding to glucocorticoid response elements in host DNA. This interferes with the antigen processing of various immune cells and inhibits the release of pro-inflammatory cytokines [16]. Although the Food and Drug Administration (FDA) has not approved topical medications for psoriasis in children younger than 12 years old, corticosteroids are the first-line treatment for psoriasis and AD in any age. In pediatric plaque psoriasis and AD, apply high-potency to medium-potency topical corticosteroids (Table 2) twice daily; once-daily treatment may also be effective. Treatment can continue daily for up to four weeks. If symptoms sufficiently improve within four weeks, discontinue topical corticosteroids or taper to intermittent use, such as the weekend. If there is no sufficient improvement within four weeks despite proper adherence, use a super-high potency topical corticosteroid for up to two weeks. After two weeks of use, taper application frequency to the weekends or transition the patient to a lower potency corticosteroid for maintenance. If flares occur, resume daily application of a high-potency corticosteroid for up to four weeks, or a super-high potency corticosteroid for up to two weeks [17].

The most common adverse events (AEs) associated with topical corticosteroids in children are cutaneous atrophy and striae. A rare but serious AE is suppression of the hypothalamic–pituitary–adrenal axis because of the systemic absorption of steroids due to the children’s increased body surface area (BSA) to volume ratio [18]. To reduce the risk of AEs, parents should avoid long-term, daily use of topical corticosteroids on large areas of skin. Using topical corticosteroids in combination and rotation with topical vitamin D analogs, topical calcineurin inhibitors, tar, and keratolytics can optimize efficacy and reduce AEs [16,17].

### 3.4. Vitamin D Analogs

Topical vitamin D analogs, such as calcipotriene and calcipotriol, are both vitamin D_3_ analogs. They are corticosteroid-sparing topical agents that can be used independently or simultaneously with topical corticosteroids for psoriasis. Vitamin D inhibits keratinocyte proliferation while promoting differentiation. During acute flares of psoriasis, patients should apply topical vitamin D analogs until sufficient improvement occurs. During maintenance therapy, use topical vitamin D analogs twice daily for five consecutive days, such as weekdays, and reserve the corticosteroid for the weekend [16]. Combination corticosteroid/topical vitamin D analogues, such as betamethasone/calcipotriol and halobetasol/calcipotriene, are available and may simplify management and improve treatment compliance. In randomized, controlled clinical trials combination topical corticosteroid/vitamin D analogues were more efficacious than either agent alone for the treatment of psoriasis [19,20,21].

### 3.5. Topical Calcineurin Inhibitors

Calcineurin inhibitors, such as tacrolimus and pimecrolimus, are corticosteroid sparing options for AD and psoriasis lesions located on the face, flexures, and intertriginous areas. Inhibition of calcineurin decreases the activity of transcription factors necessary for cell division. This selectively prevents T-cell activation and provides an anti-inflammatory effect [22]. Tacrolimus 0.03% ointment is approved for AD in children 2 to 15 years old. Tacrolimus ointment, applied to the face or intertriginous areas twice daily, is an effective treatment for AD. Tacrolimus reduces clinical AD symptoms in as little as 1 week and remains efficacious and safe for up to 4 years [23,24]. Similarly, a study of pimecrolimus use in infants with mild to moderate AD showed rapid efficacy with good long term safety. Over 50% of subjects achieved treatment success within 3 weeks. After 5 years, greater than 85% achieved overall and facial treatment success [25]. Tacrolimus has also been tried off label in pediatric psoriasis. In a retrospective study of pediatric inverse psoriasis, 12 of 13 patients had complete clearance of lesions within 2 weeks of starting topical therapy with 0.1% tacrolimus [26]. Proactive application of topical calcineurin inhibitors may prevent flares of AD and psoriasis and reduce total amount of steroid use [27,28,29]. Temporary site pain, burning, or stinging, is the most common AE. Despite no evidence suggesting an increased risk of lymphoma and skin cancers, avoidance of excessive sun exposure and phototherapy is recommended for patients undergoing calcineurin treatment [16,17].

### 3.6. Topical Phosphodiesterase-4 Inhibitor

Crisaborole, a novel phosphodiesterase (PDE)-4 inhibitor, was FDA approved in 2016 for use in mild-to-moderate AD patients 2 years of age and up. In a 2017 double blind study, more patients treated with crisaborole achieved a reduction in the Investigator’s Static Global Assessment (IGSA) score and improvement in pruritus at day 29 compared to those treated with ointment vehicle only [30]. Crisaborole 2% can be applied twice daily to affected areas with no limitation on duration of use. The most commonly reported AE was application site pain in a minority of patients. It is considered an effective and generally well tolerated treatment option [31,32]. However, no head-to-head studies versus established treatments have been published to define true efficacy.

### 3.7. Oral Therapy

Topical treatments and daily skin care are considered first line therapy for AD and psoriasis. Patients with moderate-to-severe or refractory disease may need escalation of care to systemic therapy (Table 3). The American Academy of Dermatology (AAD) recommends systemic therapy when signs and symptoms of the disease are not adequately controlled with optimal topical regimens or when the patient’s skin disease is so severe that it significantly impacts physical, emotional, or social wellbeing [33,34]. The American Academy of Dermatology (AAD) guidelines also state that once disease control has been attained, the lowest effective dose and duration of systemic therapy should be used. Adjunctive therapies such as topicals should be continued to help meet this goal. If the patient has a large body surface area affected, topical applications can be difficult, expensive, and time consuming, which may lead to decreased adherence. Systemic treatments have their own hurdles to adherence such as routine lab monitoring and scheduled dosing regimens.

### 3.8. Cyclosporine

Cyclosporine (CsA) is a potent immunosuppressant that inhibits interleukin-2 (IL-2) production and disrupts T-cell mediated functions. Due to its unfavorable side effect profile, it is generally used for short courses during a flare or a bridge between therapies. For AD treatment, cyclosporine is usually started at 5 mg/kg/day followed by tapering. In a retrospective study of children with severe AD, 64% reported an excellent or good outcome after 4 weeks of treatment [35]. A prospective study of pediatric psoriasis therapy reported a 75% reduction of psoriasis area and severity index (PASI) in 40% of its patients treated with CsA [36].

CsA is reserved for unstable, severe, recalcitrant, erythrodermic, or palmoplantar psoriasis. CsA is FDA-approved for adults with severe psoriasis and pediatric transplant patients more than 6 months old. The efficacy of CsA in pediatric psoriasis is supported by only limited data [37]. Treatment doses should be 1.5–5 mg/kg/day [38]. Due to pharmacokinetic differences between children and adults, pediatric patients may require higher doses of CsA. The differences that explain this higher dosing requirement in children include lower oral absorption, more rapid clearance, and greater volume distribution [39]. Clinical improvement is seen 4–8 weeks after starting treatment [40,41]. Once improvement is achieved, taper the dose to the lowest effective maintenance dose. Treatment with CsA should be less than one year to limit AEs, as CsA is associated with high rates of severe AEs [41,42]. Serious AEs include hypertension, nephrotoxicity, hepatotoxicity, hyperlipidemia, metabolic abnormalities, and increased risk for infections and malignancy, such as skin cancer and lymphoproliferative disorders. Avoid combining CsA with phototherapy due to the increased long-term risk of skin cancer [43]. CsA requires routine blood pressure checks and laboratory monitoring for hematologic and hepatic AEs. Children treated with CsA should avoid live vaccines and macrolide antibiotics, which can increase CsA levels.

### 3.9. Methotrexate

Methotrexate (MTX) is considered an off-label, first-line systemic therapy for children with severe, erythrodermic, or recalcitrant psoriasis [42]. MTX can also be used for refractory AD [44]. Clinical improvement occurred in 76% of 55 moderate to severe AD pediatric patients with a decrease of mean Investigator’s Global Assessment from 4.2 to 3.0 after 6 to 9 months of MTX treatment. There was additional improvement after 12 to 15 months [45]. MTX provided a reduction of Psoriasis Area and Severity Index (PASI) by 50% or greater in 58% in a small prospective study [36]. Advantages of MTX include a long history of use in pediatric populations, low cost, and oral formulation.

Doses should be 0.2–0.7 mg/kg/weekly, with a maximum dose of 25 mg/week [46]. Once improvement is achieved, taper MTX to achieve the lowest effective dose for maintenance. If a stable dose controls disease for six months, taper 0.1 mg/kg each month. Common AEs of MTX include gastrointestinal upset, upper respiratory infections, fatigue, and headaches. Serious AEs include bone marrow suppression, transaminase abnormalities, and MTX-induced lung disease. Folic acid 1 mg/day together with MTX minimizes AEs. Use of MTX requires routine laboratory monitoring for hematologic and hepatic AEs. Children should avoid live vaccines and sulfa drugs while receiving MTX. Children need to tolerate laboratory monitoring, be up to date on vaccinations, and lack risk factors for hepatoxicity. Females of childbearing age should not use MTX without contraceptive therapy due to risk of teratogenicity and fetotoxicity. Maximal effect typically requires 2–3 months of therapy to achieve [33,34].

### 3.10. Azathioprine

Azathioprine (Az) is an inhibitor of purine synthesis that reduces leukocyte proliferation and can be used for severe or refractory AD [47,48]. A small study reported clinical improvement, without quantification, in 11 of 12 pediatric patients with recalcitrant AD treated with Az and few AEs [47]. The catabolism and active metabolite production of Az is carried out by the enzyme thiopurine methyltransferase (TPMT). The level of activity of TPMT is regulated by common allelic polymorphisms, thus producing variability from patient-to-patient [33]. Therapeutic dosing can be guided by genotyping for TPMT polymorphisms or enzyme activity level. If enzyme activity is high, dosing should be 2–3.5 mg/kg/day. If enzyme activity is low, dosing should be 0.5–1.0 mg/kg/day. The onset of action is slow with clinical improvement in 6 to 8 weeks and 12 weeks for an adequate trial [34,48].

The principle AEs associated with Az are myelosuppression, hepatotoxicity, gastrointestinal disturbance, and increased risk of malignancy especially with prolonged use. Monitoring of liver function and hematologic parameters is required. Measurement of TPMT activity level decreases the risk of complications [33].

### 3.11. Mycophenolate Mofetil

Mycophenolate mofetil (MMF) inhibits the de novo synthesis of purine nucleotides thereby preventing proliferation of lymphocytes. MMF can be considered as a therapeutic option for moderate to severe or refractory AD and psoriasis [49,50,51]. In a small retrospective study of 14 pediatric AD patients, 13 had a 60% or greater improvement, with 4 patients achieving complete clearance [50]. Pediatric dosing is 30–50 mg/kg/day. Gastrointestinal disturbance and dose dependent myelosuppression are the most common AEs [33].

### 3.12. Acitretin and Other Vitamin A derivatives

Acitretin and etretinate are second generation synthetic retinoids, otherwise known as aromatic retinoids [52]. Acitretin is useful in recalcitrant psoriasis in pediatric populations when psoriasis refractory to topical agents and phototherapy significantly impairs the life quality of the affected child; acitretin is most effective in the pustular and erythrodermic subtypes of psoriasis, and slightly less efficacious in plaque psoriasis. Vitamin A analogs offer the advantage of being a non-immunosuppressive systemic medication and can be used even in children who have contraindications to immunosuppressant therapy [52]. Acitretin also has the advantage that it can be used as combination therapy with phototherapy; acitretin has a synergistic effect with narrow band UVB (NB-UVB) phototherapy [52]. One major limiting factor of acitretin use in the pediatric population is concern for skeletal toxicity in children, although this risk can be minimized by using the minimum effective dose (0.5–1 mg/kg/day), proper and regular physical exams, and laboratory tests. Acitretin can treat pediatric pustular, erythrodermic psoriasis without causing serious radiologic abnormalities [52,53].

## 4. Discussion

Management of pediatric AD and psoriasis can include both topical and systemic medication. If AD or psoriasis is moderate-to-severe, or resistant to topical treatments, escalation to systemic oral medications may be necessary. Systemic medications carry a risk of adverse effects, which may be concerning to patients and caregivers. Therefore, if a patient appears to have resistant disease, it is worthwhile to assess adherence before escalating care. Topical and systemic medications are effective when used properly. However, AD and psoriasis are chronic conditions that require long term continuous management plans which can be problematic for patients to follow and lack of adherence is commonly reported. Application of topicals can be time consuming and messy. Patients can become frustrated with perceived lack of efficacy or fear of side effects. Commonly employed treatment strategies include; using the simplest yet most effective plan possible, including the patient in treatment plan decision, patient education, and involving friends or family members in the patient’s medication application. Increasing proper medication adherence in children results in better health outcomes and better management of their disease as they approach adulthood.

Limitations of this study include the scarcity of standardized guidelines regarding treatment choice, dosing and long-term safety for childhood atopic dermatitis and plaque psoriasis. As this is a classical review based on one database, there is inherent ascertainment bias when selecting articles for inclusion. This is not a meta-analysis and there are few head-to-head trials which to compare the various topical and oral therapies. Therefore, it is difficult to create fixed treatment guidelines.

Current guidelines for management of AD are based on the patient’s severity of disease [6]. Non-pharmacological treatment with regular moisturizer or emollient use and proper cleansing and bathing is a fundamental part of treatment [7,16]. Non-pharmacological approaches can reduce severity of AD and limit the need for pharmacological treatment. Topical corticosteroids are the preferred treatment for individuals who fail to respond to non-pharmacological treatment. There are various strengths and preparations of topical corticosteroids. There are a variety of factors that are taken into consideration when selecting a topical corticosteroid for AD patients, including age of patient, area to which the topical will be applied, cost of medication, and patient preference. To minimize risk of adverse events, potential options include selecting the lowest strength topical corticosteroid that is effective or selecting a stronger potency topical corticosteroid for a shorter period of time [7,16]. Topical calcineurin inhibitors such as tacrolimus and pimecrolimus and the topical PDE-4 inhibitor crisaborole are steroid-sparing topical agents that can be used in AD patients recalcitrant to topical corticosteroids or those with long-term, uninterrupted topical corticosteroid use. They can be used as a single agent or in combination with topical corticosteroids for acute or chronic AD [16].

Phototherapy, when feasible, is a second line treatment option for AD after failure with first line topical agents. The dosing and frequency of light therapy is generally based on Fitzpatrick skin type and minimal erythema dose [16]. Phototherapy is a viable maintenance treatment option for chronic AD. Systemic agents such as cyclosporine, azathioprine, methotrexate, mycophenolate mofetil, and acitretin are immunomodulatory agents used as treatment for AD in patients who fail topical regimens and/or phototherapy. These immunomodulatory agents have varying adverse effects and treatment decisions factor in the individual’s AD severity, preferences, and comorbid conditions. The complexity of the AD guidelines (bathing, moisturizers, topical medications) poses a challenge to treatment adherence. Written AD action plans, often called eczema action plans (EAPs) can be used to improve patient and caregiver understanding of treatment regimens [54,55]. These plans give detailed instructions for bathing, moisturizing, and specific medication use depending whether the patient’s AD is under control or flaring. However, even with the addition of written action plans, adherence remains a major hurdle for AD treatment.

Treatment guidelines for psoriasis are based on extent of disease, commonly classified by body surface area involvement [17]. Those with limited disease (<5% body surface area involvement) can be treated with topical agents which are generally safe and efficacious. Topical corticosteroids are the mainstay of topical treatment for mild to moderate plaque psoriasis. Patient age, location being treated, disease severity, and patient preference impact choice of potency and vehicle of the topical corticosteroid. Vitamin D analogues and calcineurin inhibitors are steroid-sparing topical agents that can be used alone or in combination with topical corticosteroids for added benefit. Coal tar has been a longstanding topical therapeutic option for psoriasis and, working through aryl-hydrocarbon receptors, has anti-inflammatory, antipruritic, antiproliferative, antibacterial and antifungal properties; the staining, irritation, and poorly tolerated odor limit its use, especially in pediatric populations. Phototherapy and systemic therapies can treat psoriasis affecting >10% body surface [34,56]. Methotrexate, cyclosporine, and acitretin are commonly used oral systemic immunomodulatory agents for psoriasis. Although efficacious, these oral agents have increased risk of toxicity and require monitoring lab work.

Future topical and oral treatments for AD and psoriasis could offer a promising alternative to standard therapy. There are several drugs—such as the small molecular oral compounds timapiprant and fevipripant—currently in phase II and III clinical trials for adult moderate to severe AD, though hopefully, if successful, these drugs will also become available to pediatric populations in the future. Timapiprant and fevipripant target the chemoattractant receptor-homologous molecule expressed on T_H_2 cells, and are both currently in phase II clinical trials for moderate to severe AD in adults.

The introduction of agents such as delgocitinib, a topical Janus kinase (JAK) inhibitor, offers efficacious treatment for AD patients with less risk of side effects compared to topical corticosteroids. However, a Phase IIa clinical study showed that efficacy with delgocitinib was not achieved until 4 weeks into treatment [57], a considerably longer period of time compared to topical corticosteroids which provide improvement of AD within days. Though only tested in adults so far, topical JAK1/JAK3 inhibitor tolfacitinib was better than vehicle alone in a phase IIa clinical trial across all endpoints (Eczema Area and Severity Index [EASI], Investigator’s Global Assessment Score [IGA], body surface area and pruritus; *p* < 0.001) [58]. Ruxolitinib (JAK1/JAK2 inhibitor) is another topical treatment currently in phase II clinical trials in children 12–17 years old and adults 16–64 [58]. It seems unlikely that new topicals will be a major advance unless they are at least as effective as potent topical steroids and completely safe; otherwise, the hurdle of abysmal adherence to topicals is unlikely to be surmounted.

There are also several oral JAK inhibitors currently in clinical trials for moderate to severe AD in both adults and children, including baricitinib (JAK1/JAK2), upadacitinib (JAK1 inhibitor), Pf-04965842 (JAK1 inhibitor), and ASN002 (JAK, tyrosine kinase 2, spleen tyrosine kinase inhibitor). For example, multiple phase III trials are evaluating the safety and efficacy of baricitinib as monotherapy in adults, and in a phase IIb clinical trial in adults, patients receiving 30 mg of upadacitinib had favorable outcomes in EASI score improvement and pruritus reduction as opposed to patients receiving only placebos [58]. Topical and oral JAK inhibitors are emerging as new treatment options for AD in both children and adults—they are more effective than placebos in clinical trials, and seem to be well-tolerated with few serious adverse effects, and may even promote rapid palliation of pruritus [58]. Since adherence to topicals for AD is appallingly low, these new oral therapies may prove to be especially promising. However, their safety is not yet well characterized, and that may limit their use.

## 5. Conclusions

Topical and systemic medications are generally effective in managing AD and psoriasis. If topical treatment is not sufficient, pediatric patients with AD or psoriasis may require escalation of care to systemic treatments, which can result in adverse effects and may require additional laboratory monitoring. However, treatment resistance may be secondary to poor treatment adherence. Therefore, assessing adherence may mitigate escalation of care and instill a sense of better disease management in patients. Because of the complexity of AD and psoriasis treatment regimens (including bathing, moisturizing, different prescription medications), providing patients with a written action plan may be beneficial; this is especially important in pediatric patients, where the patients are oftentimes reliant on their caregiver’s understanding of their condition and its proper treatment. Such a written action plan can provide patients and their families information on when and how often to bathe, moisturize, and apply or take their medication, how much of the product to use and where they should apply it, different instructions depending on the condition of the patient’s skin (maintenance care, mild/moderate flare, severe flare), as well as a phone number on which they can reach their provider if they have questions or concerns [59].

Maintaining adherence of patients with AD and psoriasis to topical treatment options remains a challenge; therefore, in addition to attempting to improve adherence to longstanding topical therapies using such tools as patient education and written action plans, novel oral and injectable drugs for AD and psoriasis currently in clinical trials may, if proven safe and effective, offer an at least partial, innovative solution to the adherence issues that are notoriously associated with topical therapies.

## Figures and Tables

**Table 1 children-06-00125-t001:** Topical treatments in pediatric atopic dermatitis and psoriasis patients.

Medication	Topical Moisturizer and Emollients	Wet Wraps	Corticosteroids	Narrow-Band UVB Phototherapy	Topical Calcineurin Inhibitors	Topical Vitamin D Analogue	Crisaborole
Use	AD Psoriasis	AD Psoriasis	AD Psoriasis	AD Psoriasis	AD Psoriasis	Psoriasis	AD
Mechanism	Improve barrier function and reduce transdermal evaporation	Provide vehicle occlusion enhancing absorption	Anti-inflammatory and anti-proliferative effects	Decreases cell proliferation, immunosuppression, T cell apoptosis	Inhibits T-Lymphocyte activation and transcription of genes that code for IL-3,4,5, GM-CSF and TNF-a	Inhibit proliferation and stimulate differentiation of keratinocytes	PDE-4 inhibitor
Dosing Regimen	At least 2x daily after bathing and handwashing	PRN	Various strengths and formulations available	Initial dose 50 of minimal erythema dose, then gradual increase to maximum tolerated dose or 2000 to 5000 mJ/cm^2^ two to five times/week	Apply a thin layer 2x daily	Apply a thin layer 2x daily avoid face and eyes	2% ointment 2x daily to affected areas
Contraindications	None	None	Local bacterial or fungal infections	Xeroderma pigmentosum, lupus erythematosus	Hypersensitivity to tacrolimus or pimecrolimus	Hypersensitivity to ingredients, hypercalcemia or vitamin D toxicity	Hypersensitivity to any ingredients
Adverse Effects	None	None	Common: striae, bruising, acne Severe: Risk of systemic absorption	Common: erythema, xerosis, pruritus, blistering Severe: increased frequency of recurrent herpes simplex, photocarcinogenesis	Burning or stinging at application site	Skin irritation	<1% contact urticaria >1%application site pain (burning or stinging)
Baseline Lab Monitoring	None	None	None	None	None	None	None
FDA-approved in pediatric populations	NA	NA	Patients >12 years of age	Patients > 6 years of age	Tacrolimus 0.03% for patients 2–15, pimecrolimus 1% for patients > 2 years of age	Not approved for pediatric patients	Patients 2 years of age and older

Abbreviations: PRN = pro re nata (as needed); NA = not available; IL = interleukin; GM-CSF = granulocyte-macrophage colony stimulating factor; TNF-α = tumor necrosis factor alpha; PDE = phosphodiesterase; FDA = Food and Drug Administration.

**Table 2 children-06-00125-t002:** Topical corticosteroids.

Drug	Preparation	Potency
Clobetasol propionate	0.05%	Super high
Budesonide	0.025%	High
Fluocinonide	0.05%	High
Triamcinolone acetonide	0.1%	Medium
Fluocinolone acetonide	0.025%	Medium
Desonide	0.05%	Low
Hydrocortisone	1%	Low

**Table 3 children-06-00125-t003:** Oral treatments in pediatric atopic dermatitis and psoriasis patients.

Medication	Cyclosporine	Methotrexate	Azathioprine	Mycophenolate	Acitretin
Use	AD Psoriasis	AD Psoriasis	AD	AD Psoriasis	Psoriasis
Mechanism	Inhibits T lymphocytes and production of IL-2 and interferon-γ	Inhibits dihydrofolate reductase	Inhibits purine synthesis	Inhibits purine synthesis	Unknown. Believed to target specific retinoid receptors in the skin which help normalize the growth cycle of skin cells
Dosing Regimen	1.5 to 5 mg/kg per day with a maximum of 5mg/kg per day. Limited to 1 year of treatment	0.1 to 0.4mg/kg/week with a maximum of 25mg/week	0.5–3.5 mg/kg/day up to 12 weeks	30–50mg/kg/day	0.2–1.0 mg/kg/day
Contraindications	Kidney disease, active infections, hypertension, malignancy, use of phototherapy	Liver disease, kidney disease, hematologic disorders immunodeficiency, pregnancy	Hypersensitivity to drug or components, pregnancy, previously treated with alkylating agents	Hypersensitivity to drug or components, pregnancy	Liver disease, kidney disease, hypertriglyceridemia, pregnancy
Adverse Effects	Common: nausea, diarrhea, arthralgia, headache Severe: nephrotoxicity, hepatotoxicity, hypertension, hyperlipidemia, gingival hyperplasia, malignancy	Common: gastrointestinal upset, upper respiratory infections, fatigue, headache Severe: bone marrow suppression, hepatotoxicity	Common: gastrointestinal upset Severe: bone marrow suppression, hepatotoxicity	Common: gastrointestinal upset Severe: bone marrow suppression	Common: dry skin, dry mouth, cheilitis, stomatitis and gingivitis and taste disturbances Severe: teratogenicity, hyperlipidemia, hepatotoxicity
Baseline Lab Monitoring	Blood pressure, CBC, CMP, Urinalysis, Lipid Profile	CBC, CMP, Lipid Profile, Hepatitis panel	CBC, CMP	CBC	CMP, CBC, lipid profile, pregnancy testing (in females of childbearing age)
FDA-approved in pediatric populations	Pediatric transplant patients >6 months of age	Treatment of juvenile idiopathic arthritis in patients >2 years of age	Treatment of renal transplantation	Treatment of renal, cardiac or hepatic transplants	Safety and efficacy in pediatric patients have not been established

Abbreviations: CBC = complete blood count; CMP = complete metabolic panel; FDA = Food and Drug Administration; IL = interleukin.

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
