# Peer review of "Topical and Oral Therapies for Childhood Atopic Dermatitis and Plaque Psoriasis"

_children, 2019, doi:10.3390/children6110125_

Round 1
Reviewer 1 Report
This paper is interesting but need a lot of corrections and changes to be included
This paper is interesting and poses a real life of the use of several topical and general treatment for the Atopic dermatitis and Plaque Psoriasis in children in the clinical daily practice.
Nevertheless, I must to do some suggestions and recommendations trying to improve this paper
1/. This is a pure classical Review, only made by Pubmed. It is not a Systematic Review neither a Scoping Review. But there are another Databases that would be interesting to consult, such as Embase, Scopus, Google Academic and so on.
Please comment the reasons for this selection and if you consider that you can improve this search, consulting 2, 3 or more databases adding to classical Pubmed.
2/. In the Abstract Section the authors comment that thay have identified 4 topical and 5 systemic treatments for these diseases. But they describe alt least 7 topical and 4 systemic . Please clarify these discrepancies.
3/. The Introduction Section is too short and contains only 8 references. Please enlarge this Section including 2-3 more paragraphs and 5-7 more references
4/. In the footnote of Table 1 please include an explanation of the acronymus PRN and NA placed at the use of wet wraps and emolients.
5/. In the Table 2 it would be convenient to change the order of the columns for easing the lectura and interpretation to the readers. Putting on the left in the first column, the name of the Drug; in the second the Preparation and in the opposite side, on the right, the Potency
6/. In the Methods Section it would be convenient to describe how was made the papers selection, how many authors have participated in this selection, the dates of publication. From haw many papers hace you started and how many were discarded. Have you used the PRISMA criteria. It would be very convenient to include in this Section, all these relevan points of the selection of the papers
7/. In the line 90, the acronymus FDA, is commonly refeared to the “Food and Drug Administration” Bureau, but not as Federal Agency
8/. In the line 173, please explain briefly the pharmacokinetic differences of Cyclosporine between children and adults.
9/. The discussion Section is too short. It contains only 2 paragraphs and no references. It would be enlarged very much inclidin 5-6 paragraphs and between 10 to 15 references
10/. In the Reference section it would be convenient to introduce several changes according with the rules of the Journal
10.1. Line 255. Ref 1. The Journal´s title must be abbreviated.
10.2. Line 257. Ref 2. The Journal´s title must be abbreviated.
10.3. Line 259. Ref. 3 . The Journal´s title must be abbreviated.
10.4. Line 262. Ref. 4. Supress pp.
10.5. Line 269. Ref. 6. The Journal´s title must be abbreviated and the year placed behind the volumen
10.6. Line 274. Ref. 7. The Journal´s title must be abbreviated and the year placed behind the volumen
10.7. Line 279. Ref. 9. The journal´s title is laking
10.8. Line 281. Ref. 10. The Journal´s title must be abbreviated.
10.9. Line 287. Ref. 12. The journal´s title is laking
10. Line 290. Ref. 13. The Journal´s title must be abbreviated.
10.11. Line 294. Ref, 14. The reference is repetead twice
10.12. Line 304. Ref. 18. The Journal´s title must be abbreviated.
10.13. Line 306. Ref. 19. The Journal´s title must be abbreviated.
10.14. Line 308. Ref. 20. The Journal´s title must be abbreviated.
10.15. Line 310. Ref. 21. The year must be moved behing the volumen
10.16. Line 314. Ref. 22. The Journal´s title must be abbreviated.
10.17. Line 317. Ref. 23. The volume and pages are lacking
10.18. Line 320. Ref. 24. The Journal´s title must be abbreviated.
10.19. Line 323. Ref. 25. The Journal´s title must be abbreviated.
10.20. Line 326. Ref 26. The Journal´s title must be abbreviated.
10.21. Line 333. Ref. 27 and 28 are duplicated (one must be eliminated)
10.22. Line 337. Ref 29. The year must be moved behind the volumen
10.23. Line 340. Ref. 30. The Journal´s title must be abbreviated.
10.24. Line 342. Ref. 31. Basel.Switzerland must be eliminated
10.25. Line 344. Ref 32 . The volumen must me placed after the year.
10.26. Line 346. Ref 33. The Journal´s title must be abbreviated.
10.27. Line 348. Ref. 34. The Journal´s title must be abbreviated.
10.28. Line 350. Ref. 35. The Journal´s title must be abbreviated.
10.29. Line 352. Ref 36. The Journal´s title must be abbreviated.
10.30. Line 355. Ref. 37. The Journal´s title must be abbreviated.
10.31. Line 357. Ref. 38. The Journal´s title must be abbreviated.
32. Line 359. Ref. 39. The volume and pages are lacking. The year must me moved and placed before the volumen.
10.33. Line 362. Ref. 40. Review the volumen because can´t be 00
10.34. Line 365. Ref. 41. The year must be placed before the volumen
10.35. Line 367. Ref. 42. The Journal´s title must be abbreviated.
10.36. Line 369. Ref. 43. The Journal´s title must be abbreviated.
10.37. Line 374. Ref. 45. The Journal´s title must be abbreviated.
10.38. Line 376. Ref 46. The month of publication and the number must be removed
10.39. Line 378. Ref. 47. The Journal´s title must be abbreviated and the yaer placed before the volume

Reviewer 2 Report
Comments to the Author
The manuscript by Travis Frantz aims to investigate “Topical and Oral Therapies for Childhood Atopic Dermatitis and Plaque Psoriasis.” However, besides the obvious retrospective design of this study.
Several typos/English spelling throughout the manuscript should be corrected. Table confusion, difficult to read and understand.
Several typos/English spelling throughout the manuscript should be corrected. Table confusion, difficult to read and understand. The tables need to be more clearly presented. Psoriasis is a chronic autoimmune condition that results in the overproduction of skin cells. The dead cells build up into silvery-white scales. The skin becomes inflamed and red, causing serious itching. Psoriasis treatment include topical (Corticosteroids), Systemic therapies (These are protein-based drugs that derive from living cells. Biologics target the T cells and immune proteins that cause psoriasis and psoriatic arthritis) Biologics are effective, but their risks is common to see. Atopic Dermatitis occurs because of a hypersensitivity reaction. This causes the skin to overreact to certain triggers, such as dyes, fabrics, soaps, animals, and other irritants Children often get atopic dermatitis (AD) during their first year of life. If a child gets AD during this time, dry and scaly patches appear on the skin. These patches often appear on the scalp, forehead, and face. These patches are very common on the cheeks. AD is often also treated with a topical corticosteroid cream. In some cases, doctors may suggest over-the-counter creams.AD no need Systemic therapies. Atopic Dermatitis several effecter T cell subsets, such as pro-inflammatory cells like Th9, Th17 and Th22 cells. Psoriasis several effecter T cell subsets, such as pro-inflammatory cells like Th23, Th17 cells. Atopic Dermatitis and Plaque Psoriasis have many similar symptoms,but treatment machine diversified. It must to be more discussed.Author Response
Please see the attachment.

Round 2
Reviewer 1 Report
I congratulate to the authors for the great effort realized and the inclusion of all the changes suggested previously
Thanks a lot for the great improvement of the paper!
